# Effects of Long-Term Application of Nitrogen Fertilizer on Soil Acidification and Biological Properties in China: A Meta-Analysis

**DOI:** 10.3390/microorganisms12081683

**Published:** 2024-08-15

**Authors:** Liqiang Zhang, Zehang Zhao, Bailing Jiang, Bate Baoyin, Zhengguo Cui, Hongyu Wang, Qiuzhu Li, Jinhu Cui

**Affiliations:** 1College of Plant Science, Jilin University, Changchun 130012, China; lqzhang23@mails.jlu.edu.cn (L.Z.); 18846915612@163.com (Z.Z.); jiangbl22@mails.jlu.edu.cn (B.J.); bybt23@mails.jlu.edu.cn (B.B.); hong_yu@jlu.edu.cn (H.W.); 2Soybean Research Institute, Jilin Academy of Agricultural Sciences, Changchun 130033, China; 17643346860@163.com

**Keywords:** nitrogen fertilizers, soil acidification, soil enzymes, soil microbial communities, functional genes, climatic conditions

## Abstract

Soil acidification is a global environmental problem with significant impacts on agricultural production, environmental protection, and ecosystem health. Soil acidification is widespread in China, affecting crop yields, agricultural product quality, and biodiversity. Since the 1980s, much work has been done on acidic soils in China, but it is controversial whether excessive nitrogen fertilizer application can lead to soil acidification mechanisms. To address the above issues, we conducted a meta-analysis of 115 published papers to integrate and analyze the effects of N fertilizer application on soil acidification and biological properties from 1980 to 2024. We also quantified the effect of nitrogen fertilization on soil acidification and biological changes under different climatic conditions. The results showed that under long-term application of nitrogen fertilizers in China from 1980 to 2024, soil pH decreased by an average of 15.27%, and the activities of soil urease, nitrate reductase, nitrite reductase, catalase, glutamate dehydrogenase, and glutamate synthetase decreased by an average of 9.82–22.37%. The soil microbial community richness (Chao1 index) increased by 6.53%, but the community diversity (Shannon index) decreased by 15.42%. Among the dominant soil microorganisms, the relative abundance of bacteria decreased by an average of 9.67–29.38% and the abundance of gene expression of *nifH*, *amoA-AOA*, *amoA-AOB*, and *qnorB* decreased by 9.92–19.83%. In addition, we found that the mean annual temperature and rainfall impacted soil acidification via their effect on soil microbial diversity and community composition. This study provides a scientific basis for an in-depth understanding of the spatial and temporal variation of soil acidification and biological properties in China.

## 1. Introduction

In the 21st century, China’s national food security issues have received increasing attention due to the scarcity of arable land resources and declining soil quality [1]. Acidic soils reduce the bioavailability of nitrogen, phosphorus, and potassium in the soil, thus affecting the uptake and use of nutrients by plants [2], and also reduce soil microbial activity, richness, and diversity, which in turn affects the function and stability of the soil ecosystem [3]. Since the 1980s, with the massive use of nitrogen fertilizers, China’s arable land has experienced significant acidification, and by 2023, China’s soil acidification area had reached 2.04 × 10^8^ ha, accounting for about 22.7% of the country’s total land area [4]. Zhu et al. (2018) found that the pH in the surface layer of 154 arable soils in 35 districts of seven provinces in China decreased by 0.5 units from 1980 to 2010 [5]. 

Soil acidification can be divided into acidification under natural conditions and acidification due to anthropogenic factors. Natural conditions include natural weathering of the soil, leaching of saline ions, decomposition of plant litter (e.g., leaves and roots) in the soil, plant root secretions, and lightning and rainfall. Anthropogenic factors include soil management practices (e.g., fertilizer application) and anthropogenic discharges (e.g., acid deposition) [6]. Among these, humic acid, produced by the decomposition of soil organic matter, is one of the most important components in acidic soils [7]. Humic acid has strong acidic functional groups, such as carboxylic and aldehyde acids, which can combine with soil particles and cations in the soil to form a stable organic matter–mineral complex. This complex has a good buffering capacity in the soil and prevents the soil pH from changing too quickly [8]. Furthermore, weathering of the parent rock, influenced by environmental factors such as temperature and precipitation, leaches the saline ionic components of the soil, resulting in desaturation of the saline base and a slightly acidic pH response of the soil [9].

Soil temperature has a large impact on soil microbial communities. Soil respiration is positively proportional to soil temperature, with a seasonal dynamic of summer > spring > autumn > winter, peaking in summer. The mechanism of action by which soil temperature affects the rate of soil respiration is through the regulation of microbial and plant respiratory enzyme activities in the soil. An increase in soil air temperature can promote the activity of plant respiratory enzymes and soil microorganisms [7]. A study showed that, when soil temperatures were within 0–35 °C, the number and diversity of microorganisms increased with increasing temperature, but when soil temperature was above 40 °C, microbial activity was significantly suppressed [10]. The number of soil microbial taxa also varies with the seasons, with bacteria being the most abundant, actinomycetes the second most abundant, and fungi the least abundant in different seasons [11]. Soil microbial populations are slightly higher in spring and summer than in autumn and winter, with bacterial populations being highest in summer and fungal populations being highest in spring. Under warmer temperatures, microbial activity is greatly increased, which favors the systematic utilization of nutrients or organic matter that may enter the soil, whereas the ability of surface or subsurface soil microorganisms to take up material is usually reduced under low temperatures [12].

Soil acidification has a direct effect on the growth and metabolic activities of soil microorganisms. Some microorganisms are more tolerant to acidic environments, but most microorganisms (e.g., *Ascomycota*) are sensitive to acidic environments, and microbial abundance and diversity tend to be lower in acidic soils [13]. A decrease in microbial abundance in acidic soils leads to a decrease in soil enzyme activity [14]. Previous studies have shown that soil pH is an important factor influencing soil enzyme activity and nutrient content [15]. Xin et al. (2024) experimentally changed the pH of a soil by increasing litter input and root biomass, which significantly improved soil enzyme activity and nutrient status. As soil microorganisms are very sensitive to changes in soil acidity and alkalinity, moderately lowering soil pH can improve soil enzyme activity, bacterial abundance, and microbial community structure, thereby optimizing the soil ecosystem and increasing soil fertility [16]. Microbial interactions also play an important role in nitrification in acidic soils. The mechanism of action is the oxidation of NO_2_^−^–N to NO_3_^−^–N, which is accomplished by nitrifying bacteria [17]. Interactions between different microorganisms can promote or inhibit the activity of nitrifying bacteria, which in turn affects the amount of nitrification. These interactions, which include mechanisms such as symbiosis, competition, and inhibition, play an important role in nitrification in acidic soils [18].

While progress has been made in understanding the effects of soil acidification, it is clear that these effects depend on specific regional environmental and soil factors. Meta-analysis is a statistical method for integrating and analyzing the results of multiple studies of the same type and obtaining consistent conclusions to deal with the heterogeneity of experimental conditions and conflicting results [19]. The aggregated information increases statistical power and provides a more comprehensive analysis of contributing factors than individual studies, ultimately producing more reliable results [20]. In recent years, the use of meta-analysis in agronomy has expanded [21,22]. The effects of time and climatic conditions on the community structure and diversity of soil microorganisms cannot be ignored. Previous studies focused on the effects of anthropogenic factors, such as agricultural practices, on the soil microenvironment, ignoring the importance of their spatial and temporal variations. In this study, we used meta-analysis to (1) clarify the effects of long-term application of nitrogen fertilizers on soil pH and biological properties; (2) determine the relationship between soil acidification, biological properties, and climatic conditions; and (3) analyze the main factors and pathways contributing to soil acidification. This study will provide a scientific basis for mitigating soil acidification and improving soil biological properties, thus providing a reference for sustainable agricultural development in China.

## 2. Materials and Methods

### 2.1. Data Collection

A meta-analysis was used to assess the effects of the long-term application of nitrogen fertilizers on soil acidification and biological properties. The Web of Science (WOS, http://apps.webofknowledge.com/, Accessed on 15 June 2024) and China National Knowledge Internet (CNKI, http://www.cnki.net/, Accessed on 24 June 2024) databases were used to find relevant peer-reviewed literature from 1980 to 2024. This was done using Boolean search terms: (“fertilization” OR “soil acidification”) AND (“soil pH” OR “soil organic matter” OR “soil enzyme” OR “soil microbial community”). Studies were screened for inclusion in the meta-analysis using the following criteria: 1) all data were obtained from open field cropping; 2) the experimental treatments consisted of both nitrogen fertilizer application (experimental group) and no nitrogen fertilizer application (control group); 3) at least one target variable was included, i.e., data on the effect of long-term nitrogen fertilizer application on soil pH, enzyme activity, or microbial diversity; and 4) the experimental treatments consisted of at least *n* = 3 replicates, for which the mean and standard deviation (SD) could be obtained from a table or from a figure using Engauge Digitizer 12.1 (https://sourceforge.net/projects/digitizer/, Accessed on 30 June 2024). If only the standard error (SE) was reported, the SD was calculated (SD = SE × √n). If neither the SD nor SE was reported, the SD was estimated as 1/10 of the mean [23]. The first screening was done by reading the titles of candidate articles and excluding any reviews and other types of literature (synthesis papers, book chapters, comments/opinions) that did not fit the scope of our meta-analysis. The second screening entailed reading the abstracts and scrutinizing the experimental trials and designs to exclude articles that failed to meet the above criteria [24]. Figure 1 shows a complete flowchart of the screening process.

### 2.2. Data Categorization

In total, 115 studies fulfilled the criteria (see Appendix A). For each study, the title, author information, literature source, trial location, climate conditions, soil classification, and agricultural management practices were extracted. Climatic conditions included the mean annual temperature (MAT) and mean annual precipitation (MAP). Although it is generally more accurate to use the precipitation and temperature observations during the crop growing season at a given site, these were not available for all studies. Therefore, MAT and MAP were used to facilitate comparison with other reports and ensure data consistency [25]. MAT and MAP were each divided into three groups (MAT: ≤15 °C, 15–20 °C, and ≥20 °C; MAP: ≤1000 mm, 1000–2000 mm, and ≥2000 mm).

### 2.3. Meta-Analysis

MetaWin 2.1.5.10 software (https://en.freedownloadmanager.org/, Accessed on 2 July 2024) was used to perform the meta-analysis. The natural logarithm of the response ratio (*R*) served as the effect size value (*lnR*), which was calculated as follows:(1)$$R=X_{t}/X_{c}$$
(2)$$lnR=ln\left(X_{t}/X_{c}\right)=lnX_{t}-lnX_{c}$$
where *X_t_* and *X_c_* are respectively the mean values of the data for the test group (N fertilizer input) and the control group (not N fertilizer input) under a saline–alkali land cropping system. *lnR* is a unitless index with a positive or negative value, respectively indicating an increase or decrease in soil pH, soil enzyme, or soil microbial community.

The variance of the effect size (*V_i_*) was calculated as follows:(3)$$V_{i}=SD_{t}^{2}/N_{t}X_{t}+SD_{c}^{2}/N_{c}X_{c}$$
where *SD_t_* and *SD_c_* are the standard deviations of the target variable in the experimental and control groups, respectively, and *N_t_* and *N_c_* are the sample sizes of the target variable in the experimental and control groups, respectively.

To achieve maximal precision, the weighted mean was used, since statistical precision surely varied considerably among the 115 studies included here. The weighted mean response rate (*lnR*++) and corresponding weights (values of *W_i_*) were calculated as follows:(4)$$lnR_{++}=\sum_{i=1}^{k}\frac{lnR_{i}\times W_{i}}{\sum_{i=1}^{k}W_{i}}$$
(5)$$W_{i}=1/V_{i}$$
where *i* and *k* denote the number of comparison and cumulative groups, respectively, and *W_i_* is the weight of a target variable’s effect size value.

The 95% CI of the effect size value was calculated via resampling (bootstrapping). If the resulting confidence interval did not overlap with zero, then the effect value was considered significant: when the entire confidence interval was greater or less than zero, this indicated N fertilizer input significantly increased or decreased the target variable, respectively (*p* < 0.05). Conversely, if the confidence interval included zero, then the N fertilizer input had no significant effect on the target variable [26]. The 95% confidence interval for *lnR++* was calculated using the following equation:(6)$$95\%CI=lnR_{++}\pm 1.96SE_{lnR_{++}}$$
(7)$$SE_{lnR_{++}}=\sqrt{\frac{1}{\sum_{i=1}^{k}W_{i}}}$$

Based on the comparison between the experimental and control groups, for descriptive purposes, their percentage change (*E*) in the target variable was also calculated, as follows:(8)$$E=\left(exp\left(lnR_{++}\right)-1\right)\times 100\%$$

### 2.4. Data Analysis

The response values of the target variables (*E*) were visualized and plotted using GraphPad Prism 9.1 software (https://www.graphpad.com/, Accessed on 2 July 2024). The relative importance of these variables for soil acidification and biological properties was explored using a random forest model in R v4.3.1 (https://www.r-project.org/, Accessed on 2 July 2024) [27]. Finally, we used structural equation modeling (SEM) to quantify the effects of the N input and environmental variables on the soil acidification and biological properties pathways [28].

## 3. Results

### 3.1. Overview of the Dataset

A final total of 115 published papers from China were included in this study (Figure 2). They contained 553 comparative soil pH observations, 2177 comparative soil enzyme (S-UE, urease; S-NR, nitrate reductase; S-NiR, nitrite reductase; S-CAT, catalase; S-SC12, sucrase; S-NPT, glutamate dehydrogenase; S-GS, protease; and S-GDH, glutamate amine synthetase) observations, 579 comparative soil microbial community alpha diversity (Shannon and Chao 1) observations, 1956 comparative soil microbial community composition (phylum level; Actinobacteriota, Proteobacteria, Acidobacteriota, Gemmatimonadetes, Ascomycota, and Basidiomycota) observations, and 765 comparative soil nitrogen cycle functional gene (*nifH*, *amoA-AOA*, *amoA-AOB*, *nirK*, nirS, *nosZ*, *qnorB*, *narG*) observations. 

Figure 3 shows the effects of long-term nitrogen fertilizer application (1980–2024) in China on soil pH, enzyme activities, abundance of dominant microorganisms, and abundance of functional genes for nitrogen cycling. The application of nitrogen fertilizers decreased the soil pH by 15.27% on average (Figure 3a). Nitrogen fertilizer also increased S-SC12 activity by 9.29%, and decreased S-UE, S-NR, S-NIR, S-CAT, S-GS, and S-GDH activities by an average of 9.82–22.37% (Figure 3a).

Long-term nitrogen fertilizer application increased the soil microbial community richness (Chao1 index) by 6.53% but decreased diversity (Shannon index) by 15.42 (Figure 3b). Furthermore, the relative abundance of Ascomycota and Basidiomycota increased by 12.83% and 20.75%, respectively. However, the application of nitrogen fertilizers decreased the relative abundance of Actinobacteriota, Proteobacteria, Acidobacteriota, and Gemmatimonadetes in the soil, with response percentages of −29.38%, −10.54%, −9.67%, and −25.37%, respectively.

As shown in Figure 3c, long-term nitrogen fertilizer application had different effects on different functional genes. The gene expression abundance of *narG* and *nirK* increased by 20.26% and 16.45%, respectively, while the gene expression abundance of *nifH*, *amoA-AOA*, *amoA-AOB*, and *qnorB* decreased by 9.92%–19.83%. The 95% confidence intervals of the response percentages of *nirS* and *nosZ* overlapped with zero, indicating that the long-term application of nitrogen fertilizer had no significant effect on the expression abundance of these two genes.

In summary, these results indicate that long-term N application has a general negative effect on soil pH and soil microbial community diversity and function.

### 3.2. Effect of Climate Conditions on the Response of Soil PH and Enzyme Activities under Long-Term Nitrogen Fertilizer Application

To further explore the effect of long-term nitrogen fertilizer on soil pH and enzyme activity, we explored the effect under different MAT and MAP categories (Figure 4a,b). The negative effect of nitrogen fertilizer on soil pH weakened with increasing MAT and MAP, from −25.46% at ≤10 °C to −3.54% at ≥20 °C, and −22.56% at ≤1000 mm to −11.43% at ≥2000 mm.

Regarding MAT, the negative effects of nitrogen fertilization on S-UE, S-NR, S-NiR, and S-CAT were strongest at 10–20 °C, with reductions ranging from 8.95% to 24.83%. Regarding MAP, the negative effects on S-UE, S-NiR, and S-CAT activities were strongest at ≥2000 mm, with a decrease of 18.29–19.89%, and the negative effect on S-NR activity was strongest at 1000–2000 mm, with a decrease of 17.89%. There was no significant effect of the long-term application of nitrogen fertilizers on S-NiR activity when MAT and MAP were at their lowest levels (≤10 °C and ≤1000 mm).

SC12 activity tended to increase with long-term nitrogen fertilization regardless of temperature and precipitation. The percentage response of S-SC12 activity was highest at ≥20 °C for MAT and at 1000–2000 mm for MAP (22.45% and 18.94%, respectively). The S-NPT, S-GS, and S-GDH activities followed similar trends to soil pH; they all responded negatively to nitrogen fertilizer, with the effects of fertilizer being weakest under MAT ≥ 20 °C and MAP ≥ 2000 mm. Moreover, under MAT ≥ 20 °C and MAP ≥ 2000 mm, there was no effect of nitrogen fertilizer on S-NPT activity.

In summary, MAT and MAP are important factors determining the effect of nitrogen fertilization on soil pH and enzyme activities, with higher MAT and MAP resulting in lower soil acidification, probably due to the increase in S-GS and S-GDH activities.

### 3.3. Effect of Climatic Conditions on the Response of Soil Microbial Community Alpha Diversity and Community Composition to Long-Term Nitrogen Fertilizer Application

The responses of soil microbial alpha diversity and taxa abundance to long-term nitrogen fertilization varied under different MAT and MAP categories (Figure 5). The positive effect of nitrogen fertilization on soil microbial community richness (Chao1 index) gradually decreased with increasing MAT, from 19.35% at ≤10 °C to 5.13% at ≥20 °C, and was highest at a MAP of 1000–2000 mm (19.24%). Microbial community diversity (Shannon’s index) responded negatively to nitrogen fertilization regardless of MAT and MAP, with the weakest effect (−14.22% and −10.66%) at the lowest MAT and MAP levels (≤10 °C and ≤1000 mm). 

In terms of microbial community composition, the positive effect of nitrogen fertilization on the relative abundance of Ascomycota and Basidiomycota was higher under higher MAT and MAP. Fertilizer had the strongest effect on the relative abundance of Basidiomycota under the highest levels of MAT and MAP (effect size: 20.05% and 19.28%, respectively). Conversely, nitrogen fertilizer application had no significant effect on the relative abundance of Basidiomycota at the lowest levels of MAT and MAP (≤10 °C and ≤1000 mm). The relative abundances of Actinobacteriota, Acidobacteriota, and Gemmatimonadetes followed similar trends. The effects of nitrogen fertilizer on these taxa were strongest at MAT ≤ 10 °C and MAP 1000–2000 mm (effect size: 11.47–29.93%). Conversely, at the highest levels of MAT and MAP (≥20 °C and ≥2000 mm), nitrogen fertilizer application had no significant effect on the relative abundance of Proteobacteria. The strongest effect of nitrogen fertilizer on the relative abundance was observed at MAT 10–20 °C and MAP ≤ 1000 mm (effect size: −14.09% and −18.54%, respectively).

In summary, the effects of nitrogen fertilization on the relative abundance of Actinobacteriota, Acidobacteriota, and Gemmatimonadetes and the diversity of soil microbial communities were strongest under higher MAT and MAP.

### 3.4. Effect of Climatic Conditions on the Response of Functional Genes Related to Nitrogen Cycling under Long-Term Nitrogen Fertilizer Application

As shown in Figure 6a,b, there were differences in the response of functional genes related to nitrogen cycling to nitrogen fertilization between different MAT and MAP categories. Nitrogen fertilizer application increased *narG* and *nirK* regardless of MAT and MAP. However, the effect on *narG* was strongest (effect size: 25.53% and 24.45%) at the lowest MAT and MAP levels (≤10 °C and ≤1000 mm, respectively). The effect on *nirK* was strongest at MAT ≥ 20 ℃ and MAP 1000–2000 mm (effect size: 19.07% and 17.99%, respectively). In addition, nitrogen application decreased *qnorB*, *amoA-AOA*, *amoA-AOB*, and *nifH* regardless of MAT and MAP, with a similar effect on each among different levels of MAP and MAT. However, the negative effect on *qnorB* and *nifH* was stronger with increasing MAT, and the effect on *amoA-AOA* and *amoA-AOB*, was weaker with increasing MAT. With respect to MAP, the effect of nitrogen on *amoA-AOA* and *amoA-AOB* was weaker with increasing MAP, but the effect of nitrogen on *qnorB* and *nifH* was strongest at a MAP of 1000–2000 mm (effect size: −19.25% and −20.95%, respectively).

### 3.5. Importance of MAT and MAP for Soil PH and Biological Properties

We used the random forest model to determine the relative importance of MAT and MAP for the soil pH and biological properties (Figure 7a–c). MAT had a greater relative effect on soil pH; S-UE, S-NR, S-CAT, and S-SC12 activities; and Shannon diversity. MAP had a greater relative effect on S-NiR, S-NPT, S-GS, and S-GDH activities and Chao 1. Moreover, MAT had a greater relative influence than MAP on the relative abundance of all taxa except Acidobacteriota and Proteobacteria. However, MAP had a greater relative influence on all nitrogen cycling genes except *nirK*. In summary, changes in MAT mainly affected soil pH and enzyme activities, while MAP affected the structure and function of soil microbial communities to a greater extent than MAT.

### 3.6. Structural Equation Modeling Analysis of the Pathways through which Long-Term Nitrogen Fertilizer Application and Climate Conditions Affect Soil Acidification

Structural equation modeling (SEM) was used to determine the relationships of N fertilizer and climate conditions with soil pH, soil enzyme activity, soil microbial community composition, alpha diversity, and N-cycle functional genes. The main pathways and modes by which N fertilizer and climate conditions affect soil acidification were determined (Figure 8). N fertilizer had a direct negative effect on soil pH (loading coefficient: −0.55). N fertilizer also reduced soil pH indirectly by reducing soil enzyme activity, soil microbial community composition, alpha diversity, and N-cycle functional genes, which in turn reduced soil pH. The climatic condition factor had a direct negative effect on soil pH (loading coefficient: −0.36). Climatic conditions also increased the composition and alpha diversity of soil microbial communities (loading coefficient: 0.46), which in turn mitigated the negative effect on soil pH. We also found that soil enzyme activities were positively correlated with soil nitrogen-cycle functional genes, and both were significantly positively correlated with soil pH. Overall, N input reduces soil pH by decreasing soil enzyme activities, microbial community composition, alpha diversity, and nitrogen-cycle functional genes, but appropriate temperature and moisture conditions can mitigate the risk of soil acidification.

## 4. Discussion

### 4.1. Effects of Long-Term Nitrogen Application on Soil PH in China

Nitrogen fertilizer has been identified as an important trigger for farmland soil acidification [29]. From the 1980s to the beginning of the 21st century, the pH of farmland soils in China declined. The contribution of nitrogen application to this decline was 70% for major crops such as wheat, maize, and rice, and 90% for fruit and vegetable crops [30]. In this study, we found that with the long-term application of nitrogen fertilizer, soil pH in China decreased by an average of 15.27% from 1980 to 2024. If soil was a closed system, the processes could be described as follows: the mineralization of molecular organic nitrogen in the soil consumes 1 H^+^ to produce 1 NH_4_^+^, 1 NH_4_^+^ nitrifies to 1 NO_3_^−^ to produce 2 H^+^, the plant absorbs 1 NO_3_^−^ to consume 1 H^+^, and the whole process of H^+^ production and excretion is balanced and does not lead to soil acidification [31,32,33]. However, soil is not a closed system, and the soil frequently exchanges substances with the atmosphere, biosphere, and hydrosphere. If the amount of nitrogen applied exceeds the amount of nitrogen taken up by plants, nitrate nitrogen will remain in the soil or be leached out, resulting in a relative decrease in nitrate nitrogen taken up by plants, a decrease in H^+^ consumed, and an increase in the amount of net H^+^, which can lead to soil acidification [34].

### 4.2. Effects of Long-Term Nitrogen Application on Soil Enzyme Activities in China

In this study, we found that long-term nitrogen application in China increased S-SC12 activity by 9.29%, whereas it decreased activities of S-UE, S-NR, S-NIR, S-CAT, S-GS, and S-GDH, with an average decrease of 9.82–22.37%. The nitrogen inputs on which plants depend are mainly nitrate and ammonium, and the nitrate absorbed by plants must be reduced to ammonium before it can be further assimilated into amino acids. This process is catalyzed by S-NR and S-NiR, while S-GS and S-GDH are mainly involved in the process of ammonia assimilation in soil [35]. S-UE, S-SC12, S-GS, and S-CAT are of vital importance for soil nitrogen accumulation and plant growth [36]. In addition, S-CAT mitigates hydrogen peroxide toxicity in crops and can promote nutrient uptake and utilization [37]. However, S-CAT activity decreased under long-term N fertilizer application, indicating that continuous N fertilizer application not only led to soil acidification but also weakened the resistance of the soil to toxicity.

### 4.3. Effects of Long-Term Nitrogen Application on the Structure and Function of Soil Microbial Communities in China

Soil microorganisms are the engine of terrestrial ecosystem functioning. Microorganisms are involved in almost all known soil matter transformation processes and are important drivers of aboveground plant productivity and nutrient use, which determine the sustainability of terrestrial ecosystems. In addition, soil microbial communities are highly sensitive to soil acidification and are important indicators of the degree of soil acidification [38]. In this study, soil microbial diversity was found to decrease by 15.42% under long-term nitrogen fertilizer application. Among the dominant microbial phyla included in this study, the relative abundance of fungi increased by an average of 12.83–20.75%, while the relative abundance of bacteria decreased by 9.67–29.38%. In addition, the expression abundance of functional genes related to nitrogen cycling such as *nifH*, *amoA-AOA*, *amoA-AOB*, and *qnorB* decreased by 9.92–19.83%. This is likely because numerous functions of soil microorganisms, such as the activities of enzymes involved in nitrogen fixation, nitrification, denitrification, and phosphorus solubilization, as well as their functional genes, are closely related to soil pH [39]. Soil acidification and its altered nutrient efficiency are key drivers of changes in the composition, structure, and diversity of soil microbial communities, leading to a decline in ecological functions in acidic soils [40]. Long-term high inputs of chemical nitrogen fertilizers are not only the main cause of acidification of agricultural soils, but also significantly alter the composition and structure of soil microbial communities, which in turn negatively affects soil microbial functions and soil health [41]. Regulations that focus on enhancing soil microbial function, such as those that improve nitrogen fixation, can reduce the need for nitrogen fertilizer application and thus mitigate soil acidification. [42]. Rare and dominant bacteria in soil microbial communities contribute differently to soil functions, with rare bacteria being more important in maintaining soil ecological functions [43]. Soil microbial keystone species play an important role in improving acid soil fertility, plant nutrient uptake, and productivity [44]. As the so-called second genome of plants, the rhizosphere microbiome has become a hot research topic in agricultural science, with studies promoting its effect on plant growth, plant stress resistance, and plant health [45]. The rhizosphere is important for the structure and function of microbial communities in acid soils under different fertilizer schemes and crops [46]. Therefore, microbial function in acid soils plays an important role in improving plant nutrition, increasing nutrient use efficiency, mitigating soil acidification, and reducing fertilizer application.

### 4.4. Climatic Conditions as a Major Influence on Soil Acidification and Changes in Biological Properties in China under Long-Term N Application

Temperature is the main factor affecting soil respiration and microbial growth, and soil temperature is the main driver of seasonal changes in microbial communities [47]. In this study, the effect of MAT on the soil microbial community was more dominant than that of MAP. This is likely due to the significant positive correlation of bacteria, fungi, and actinomycetes with soil temperature. Soil microbial abundance increased gradually from spring to summer, peaked in summer, and then declined; bacterial abundance peaked in July and fungal diversity in September [48]. Seasonal freezing and thawing affect 70% of the global land area. Soil microbial activity and community composition are strongly responsive to seasonal freezing and thawing; a sharp decrease in temperature leads to structural changes in the soil microbial community, resulting in a decrease in the number, abundance, and diversity of bacterial taxa but a significant increase in the number of fungal species and fungal/bacterial ratios. Moreover, the freezing and thawing cycle has a strong effect on the microbial metabolic community, which is predominantly bacterial in summer and fungal-dominated in winter [49]. Dominant populations at higher temperatures can metabolize substrates not used by members of the microbial community at lower temperatures. Moreover, the freezing process can directly or indirectly kill microorganisms by limiting the availability of soil water and nutrients, thereby reducing microbial abundance, with fungi being more resistant to low temperatures. Surviving species have greater tolerances and adaptations to low temperatures [50]. The main factors influencing the seasonal variation of microorganisms vary due to differences in dominant vegetation types and climatic hydrothermal conditions in different regions, with precipitation having the greatest influence on microbial abundance in the tropics, whereas air temperature plays a dominant role in the subtropics [51].

Soil water content is one of the dominant factors affecting soil microbial activities, and soil microbial basal respiration is closely related to precipitation [52]. In this study, the positive effect of nitrogen fertilizer application on soil microbial community richness was strongest at a MAP of 1000–2000 mm. However, excessive moisture can reduce the permeability of wetland soils, impede the diffusion and circulation of gases in the soil, and lead to a decrease in the abundance of soil microorganisms [53]. Changes in MAP can reduce the diversity of soil microbial communities and shift them towards more fungal-dominated systems [54]. Seasonal and temporal variations in rainfall, particularly in ecosystems where organisms may be at or near the limits of their physiological tolerance, can have a major impact on the diversity, abundance, and responsiveness of soil microbial communities. In drylands, fungal abundance is particularly low during the pre-dry season and increases significantly during the rainy season, when soil moisture is 4.7 times higher than in drylands, and bacteria are more affected by short-term changes in rainfall [55].

## 5. Conclusions

Soil acidification has become one of the most serious land degradation problems in global agricultural systems. By analyzing 4864 sets of data comparisons from 115 studies, this meta-analysis revealed that under long-term N fertilizer application from 1980 to 2024, soil pH decreased by 15.27%, soil enzyme activities decreased by 9.82–22.37%, the relative abundance of bacteria decreased by an average of 9.67–29.38%, and the gene expression abundance of *nifH*, *amoA-AOA*, *amoA-AOB*, and *qnorB* decreased by 9.92–19.83%. We also found that the mean annual temperature and rainfall could affect the rate of soil acidification via changes to the diversity and composition of soil microbial communities. To reduce the effect of nitrogen fertilizer on soil acidification, we recommend the development of efficient, environmentally friendly, and low-cost slow/controlled-release fertilizers. These findings improve our understanding of soil pH–microbe interactions. This study provides insights for the sustainable development of Chinese agriculture.

## Figures and Tables

**Figure 1 microorganisms-12-01683-f001:**
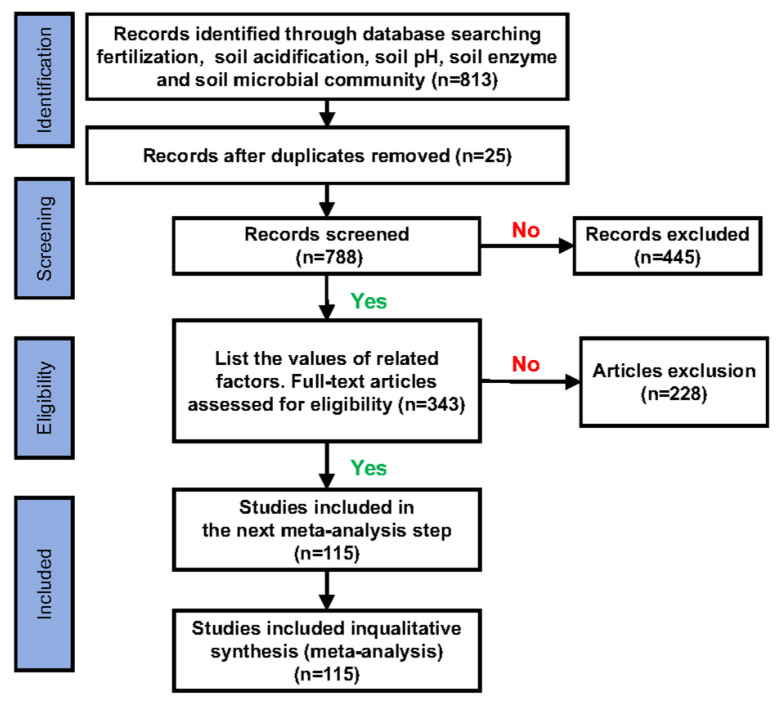
Flowchart showing the screening process for study inclusion in the meta-analysis.

**Figure 2 microorganisms-12-01683-f002:**
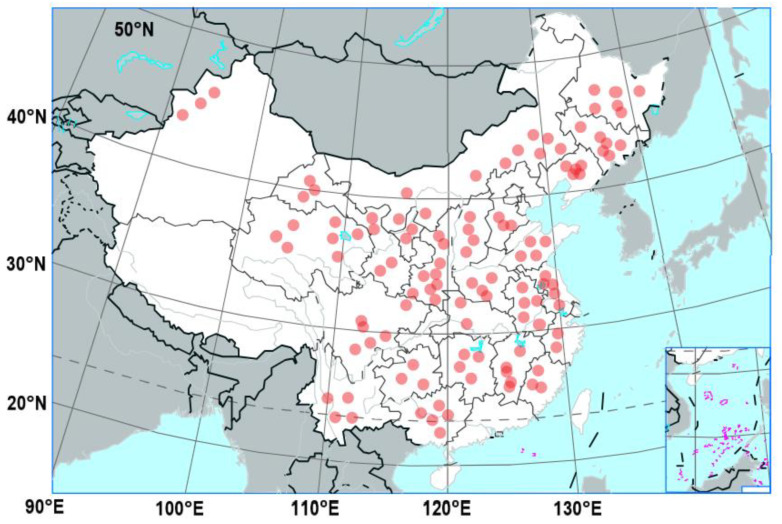
Spatial distribution of the published papers.

**Figure 3 microorganisms-12-01683-f003:**
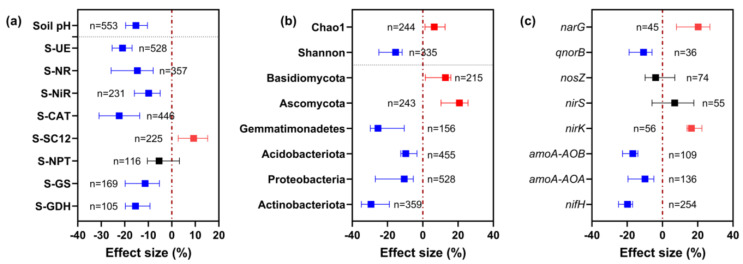
Effects of long-term N application on soil pH and enzyme activity (**a**), microbial community diversity and taxa abundance (**b**), and microbial functional gene abundance (**c**). Symbols and error bars indicate effect sizes and 95% confidence intervals (CIs), respectively. N fertilization treatments were significantly different from no-N-fertilization treatments when the CIs did not overlap with zero (red dashed line) (*p* < 0.05). Red symbols indicate an increasing effect, blue symbols a decreasing effect, and black symbols no significant effect. n is the number of samples for the variable of interest. S-UE, urease; S-NR, nitrate reductase; S-NiR, nitrite reductase; S-CAT, catalase; S-SC12, sucrase; S-NPT, glutamate dehydrogenase; S-GS, protease; S-GDH, glutamate amine synthetase.

**Figure 4 microorganisms-12-01683-f004:**
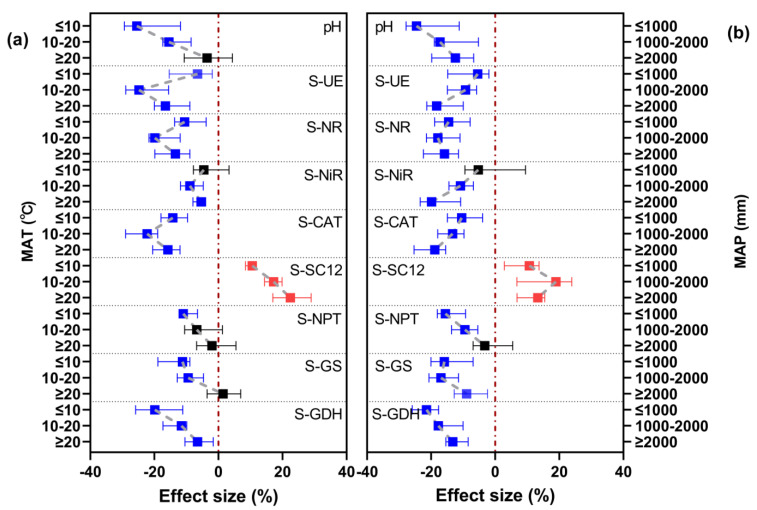
Effects of long-term N fertilization on soil pH and enzyme activities under different (**a**) MAT and (**b**) MAP categories. Symbols and error bars indicate effect sizes and 95% confidence intervals (CIs), respectively. N fertilization treatments were significantly different from no-N-fertilization treatments when the CIs did not overlap with zero (red dashed line) (*p* < 0.05). Red symbols indicate an increasing effect, blue symbols a decreasing effect, and black symbols no significant effect. S-UE, urease; S-NR, nitrate reductase; S-NiR, nitrite reductase; S-CAT, catalase; S-SC12, sucrase; S-NPT, glutamate dehydrogenase; S-GS, protease; S-GDH, glutamate amine synthetase.

**Figure 5 microorganisms-12-01683-f005:**
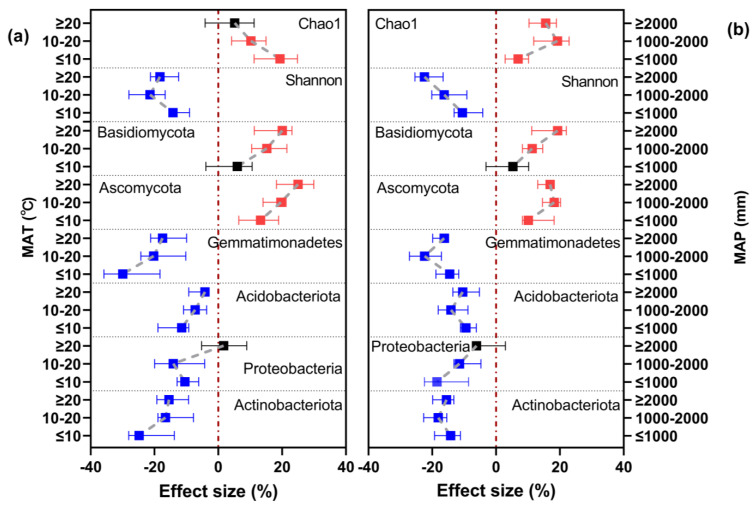
Effects of long-term N fertilization on alpha diversity and community composition of soil microbial communities under different (**a**) MAT and (**b**) MAP categories. Symbols and error bars respectively indicate the values of the effect sizes and their 95% confidence intervals (CIs). N fertilization treatments were significantly different from no-N-fertilization treatments when the CIs did not overlap with zero (red dashed line) (*p* < 0.05), with red symbols indicating an increasing effect, blue symbols a decreasing effect, and black symbols no significant effect. MAT, mean annual temperature; MAP, mean annual precipitation.

**Figure 6 microorganisms-12-01683-f006:**
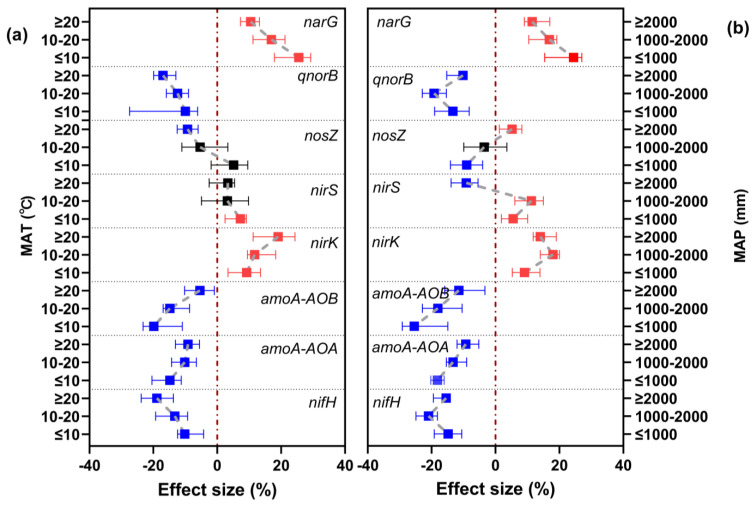
Effects of long-term N fertilization on the abundance of functional genes related to soil nitrogen cycling under different (**a**) MAT and (**b**) MAP categories. Symbols and error bars respectively indicate effect sizes and their 95% confidence intervals (CIs). Nitrogen application treatments were significantly different from no-N-application treatments when the CIs did not overlap with zero (red dashed line) (*p* < 0.05). Red symbols indicate increasing effects, blue symbols indicate decreasing effects, and black symbols indicate no significant effects. MAT, mean annual temperature; MAP, mean annual precipitation.

**Figure 7 microorganisms-12-01683-f007:**
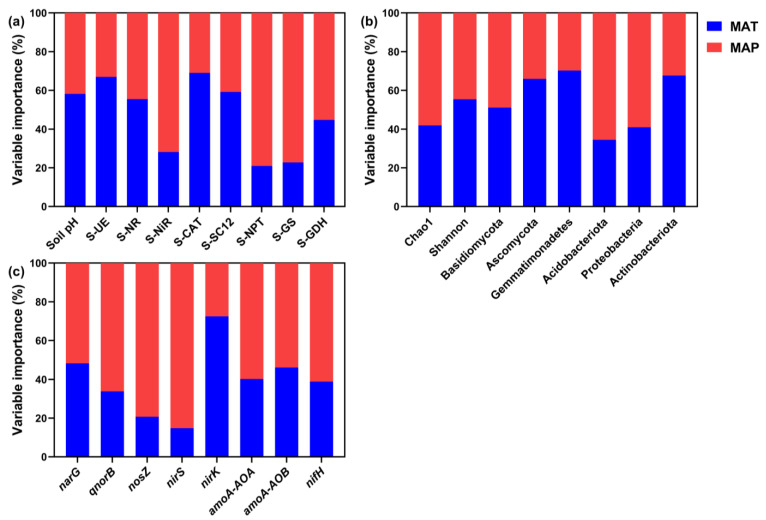
The relative influence of MAT and MAP on soil pH and enzyme activities (**a**), soil microbial community alpha diversity and community composition (**b**), and soil nitrogen cycling functional genes (**c**) based on the random forest model. Red indicates MAP and blue indicates MAT. MAP, mean annual precipitation; MAT, mean annual temperature; S-UE, urease; S-NR, nitrate reductase; S-NiR, nitrite reductase; S-CAT, catalase; S-SC12, sucrase; S-NPT, glutamate dehydrogenase; S-GS, protease; S-GDH, glutamate amine synthetase.

**Figure 8 microorganisms-12-01683-f008:**
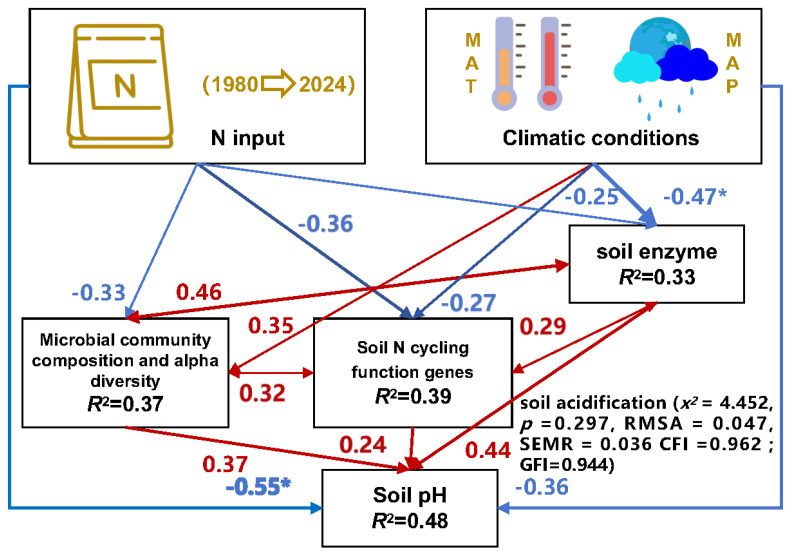
Structural equation modeling (SEM) describing the effects of N input, climatic conditions, and related factors (soil enzyme activity, soil microbial community composition, alpha diversity, and soil N-cycle functional genes) on soil acidification. Blue and red indicate negative and positive correlations, respectively (* *p* < 0.05). MAT, mean annual temperature; MAP, mean annual precipitation.

## Data Availability

The datasets generated and analyzed during the current study are available from the corresponding author upon reasonable request.

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
