# Peer review of "Effects of Long-Term Application of Nitrogen Fertilizer on Soil Acidification and Biological Properties in China: A Meta-Analysis"

_microorganisms, 2024, doi:10.3390/microorganisms12081683_

Round 1

Reviewer 1 Report

Comments and Suggestions for Authors

Thank you for the opportunity to review your manuscript “Effects of long-term application of nitrogen fertilizer on soil acidification and biological properties: a meta-analysis in China” The paper is generally well-written and addresses an important topic. However, I have a few suggestions that could help improve the clarity and impact of your study:

-In many sentences, space between the end and start of a new sentence is needed. Check the line 16. I highlighted yellow. 

-Some sentences are overly long and redundant. I suggest that the authors rewrite these sentences for better readability and conciseness. Long sentences need to be split into small lines sentence for better understanding of readers. 

-Removing unnecessary words and improving sentence structure will help streamline the text.

-What is the meaning of apoplastic matter, give the example (Line 55).

-Very confusing results (Response of climatic conditions to soil pH and enzyme activities under long-term nitrogen fertiliser application). Explain the results by splitting them into small paragraphs. It will help the reader to understand the results better. 

-Remove the extra punctuation like you used commas several times in a single sentence. 

-Some sentences need more and more recent references in the discussion, I highlighted and provided references there.

-The conclusion needs to improve, The author straightly wrote the results that are already discussed in the abstract and results & discussion section...You don’t need to do this.

-The authors need to give the significance and impact of this study. How does your result vary or be significant from already available results? How your results or study will fill the gap existing between fertilizer application and soil acidification etc?  

Reviewer 2 Report

Comments and Suggestions for Authors

Abstract.

Line 18, Please mention here the number corresponding to the filtered papers.

Introduction.

Line 37. Please add “China’s” before “national”

Line 38. Please remove “Acid in”

Line 39. Please substitute “effectiveness” with “bioavailability”

Line 40. Please substitute “Second, acidic soil also” with “This condition”

Line 41. Please remove the word “beneficial”

Line 48. Please remove the word "significantly”

Lines 49-51. Please remove from “Darilek” to “microbial diversity”

Line 78. Why are fungal and actinomycetes mentioned as different groups?

Line 78. Please define “suitable temperatures” or substitute with “warmer”

Line 86. Please mention examples.

Lines 92-95. Please cite “As soil microorganisms are… …increasing soil fertility”

Lines 95-96. Please mention the mechanisms proposed for “Microbial interactions also play an important role in nitrification in acidic soils” (besides ecological guilds).

Line 101. Please remove the word “well”

Line 101. Please substitute “in” with “as”

Lines 102-103. Please substitute “but the experimental methods are different, the results of the studies are widely different, and the relevant” with “however,”

Lines 104-105. Please remove “which cannot explain the contradictions between large-scale datasets and regions”

Line 105. Please substitute “In contrast” with “To ordain existing evidence”

Line 113. Please remove the word “community”

Materials and methods

Line 136. Please substitute “literature” with “considered works”

Lines 131-140. Please include as supplementary material a list of all papers (813) (full cite and DOI if applicable) indicating those cases that were not considered in the analysis (698).

Line 149. Please remove the acronym.

Line 153-155. For the 115 articles considered in the analysis, please present a supplementary Excel table including all the crude data extracted, indicating those cases where Engauge Digitizer was employed.

Table 1. Please remove and mention in the text. Include the classification for MAT and MAP in the Excel table mentioned in the previous comment.

Line 181. Please correct the inconsistency between the 115 studies mentioned in Figure 1 and 113 mentioned in this section.

Results

Figure 2. Please remove the box in the lower right corner.

Line 227. Please explain what “within the Chinese scale” refers to.

Lone 228. Please add “data in” after Figure 3b

Figures 3, 4, 5, and 6. Please present data as violin boxes showing data points for each measure. Please add statistical significance asterisks or letters for each category analyzed.

Figure 7. Please remove it from the main text and present it as a supplementary figure.

Discussion

Line 397. Please substitute “will decrease” with “decreased”

Line 398. Please remove “some scholars believe that urea is an alkaline fertiliser, and based on the principle of acid-base balance, it is believed that the application of urea to the soil will not lead to soil acidification”

Line 448. Please add “so-called” before “second”

Comments on the Quality of English Language

English needs a professional style revision. Grammar, orthographic, and formal language errors were found in all sections.

Reviewer 3 Report

Comments and Suggestions for Authors

Research is relevant and interesting.

1. The authors could provide their summary or suggestion at the end of each section;

2. I advise splitting the discussion into smaller chapters;

3. The conclusions could be more concrete with a proposal for the future.

Reviewer 4 Report

Comments and Suggestions for Authors

Interesting work that highlights the long-term effects of a certain agricultural management, such as nitrogen fertilization, and the modulators of this expression. I would like to make the following reflections to the authors:

In L131, the authors indicate the criteria for the selection of papers selected for meta-analysis, and the first of these is that all data should correspond to open field cropping. However, in L159-162, they indicate that they have used data from greenhouse studies. Therefore, it would be correct to state that studies have been selected that do not meet criterion 1 established by the authors? In this case, how many greenhouse studies have been used?

On the other hand, it is presumably a typographical error to indicate in L181 that 113 studies have been used, instead of the 115 indicated by the authors throughout the study. The error should be corrected.

The authors indicate, L398, that the main nitrogen fertilizer used in China is urea. It would be interesting for the authors to refer to the type(s) of nitrogen fertilizer used in the selected works and its possible impact on the parameters analyzed.

Finally, it would be appropriate for the authors to provide references to the selected studies, although not in the main text of the paper, but as complementary material or a database with access to it.

Many thanks to the authors for their work and I hope they will take my suggestions into consideration.

Round 2

Reviewer 2 Report

Comments and Suggestions for Authors

The authors did not attend relevant previous recommendations.

The key reccomendation were: 

1) To present all the literature database considered (not only the final 115 works)

2) To present data in figures 3 to 6 showing the individual data points at each condition as violin diagrams with outliners. The average and error bars are weak indicators.

3) Figures are pixelated.

Author Response

1) To present all the literature database considered (not only the final 115 works)

Thank you, we are unable to provide this as we have no record of the papers that have been excluded, but the sources and raw data for 115 papers have been submitted to the submission system.

2) To present data in figures 3 to 6 showing the individual data points at each condition as violin diagrams with outliners. The average and error bars are weak indicators.

Thank you, we disagree with the reviewer's statement, e.g., "The average and error bars are weak indicators." Here are some similar published papers that I referenced, 1. http://dx.doi.org/10.1016/j .scitotenv.2023.163226 2. http://dx.doi.org/10.1016/j.scitotenv.2023.163226 3. https://doi.org/10.1016/j.fcr.2023.109218 None of these papers include the reviewer's proposed these, and furthermore, the process of analysing these data, which we have flagged in the supplementary material. Let me ask the reviewers here, more than 500 points converging inside a few violin plots is very confusing, and it really does not bother the readers?

3) Figures are pixelated.

The images in the manuscript are currently 1200 dpi, far exceeding the 300 dpi required in the author's submission instructions.

Reviewer 4 Report

Comments and Suggestions for Authors

I believe that your work is of great interest, as it can serve as a reference tool in decision-making regarding the balance between the environment and socio-economic development. I encourage you to continue deepening these studies.

Author Response

Reviewer:I believe that your work is of great interest, as it can serve as a reference tool in decision-making regarding the balance between the environment and socio-economic development. I encourage you to continue deepening these studies.

We thank the reviewers for their support and recognition of our work, and our team will continue to refine the